# Blind Recognition of Convolutional Codes Based on the ConvLSTM Temporal Feature Network

**DOI:** 10.3390/s25041000

**Published:** 2025-02-07

**Authors:** Lu Xu, Yixin Ma, Rui Shi, Juanjuan Li, Yijia Zhang

**Affiliations:** School of Information Science and Technology, Zhejiang Sci-Tech University, Hangzhou 310018, China; xulu@zstu.edu.cn (L.X.); 202220702019@mails.zstu.edu.cn (Y.M.); 2023220704058@mails.zstu.edu.cn (R.S.); 202230704069@mails.zstu.edu.cn (J.L.)

**Keywords:** channel-coding recognition, convolutional codes, wireless communication, temporal feature network, deep learning

## Abstract

The accurate identification of channel-coding types plays a crucial role in wireless communication systems. The recognition of convolutional codes presents challenges, primarily due to their strong temporal dependencies, varying constraint lengths, and additional contamination from noise. However, existing algorithms often rely on manual feature extraction or are limited to a restricted number of coding types, rendering them inadequate for practical applications. To tackle this problem, we propose ConvLSTM-TFN (temporal feature network), an innovative blind-recognition network that integrates convolutional layers, long short-term memory (LSTM) networks, and a self-attention mechanism. The proposed approach enhances the acquisition of features from soft-decision sequence information, leading to improved recognition performance without necessitating prior knowledge of coding parameters, sequence starting positions, or other metadata. The experimental results demonstrate that our method is effective within a signal-to-noise ratio (SNR) range of 0 to 20 dB, achieving more than 90% recognition accuracy across 17 convolutional code types, with an average accuracy of 98.7%. Our method effectively distinguishes diverse coding features, surpassing existing models and establishing a new benchmark for channel-coding recognition.

## 1. Introduction

In wireless communication systems, signals are transmitted from the source and received by a receiver sensor after passing through a communication channel. However, the presence of diverse forms of noise and fading in the channel introduces distortions, leading to disparities between the received signal and the originally transmitted signal. To address this issue, channel-coding techniques have been developed to detect and rectify errors in the transmitted data.

Convolutional codes, constituting one of the most crucial and extensive channel-coding schemes in digital communication systems [1], assume a pivotal role in enhancing data transmission reliability over noisy and fading channels. Due to the indispensable nature of channel coding, the blind recognition of channel codes has found extensive application in fields such as electronic countermeasures [2,3] and adaptive modulation and coding (AMC) [4,5,6]. Especially in non-cooperative scenarios, such as in electronic warfare, military, spectrum monitoring, signals intelligence, and communications intelligence systems, the receiver possesses limited knowledge regarding the coding type and parameters used by the sender [7]. The blind estimation of the interception sequence is therefore necessary to determine the coding type and parameters accurately.

The transmission of convolutional codes is performed in a serial manner, enabling a more efficient utilization of storage space during the encoding process. However, the theoretical analysis of convolutional codes poses significantly greater challenges compared to linear block codes. Convolutional codes typically possess good error-correction capabilities, but error propagation may occur if the decoder incorrectly selects a wrong code or decoding path. The complexities highlighted here emphasize the significance of the precise recognition of convolutional codes. Convolutional code recognition and parameter estimation methods have predominantly relied on the Galois field GF(2). Hard decision-based convolutional code recognition methods have been extensively developed, enabling the recognition of various convolutional codes without necessitating prior knowledge [8,9]. Nevertheless, these approaches generally demonstrate limited resilience to noise.

Due to the suboptimal performance of convolutional code recognition under a low signal-to-noise ratio (SNR), there has been a shift towards parameter identification of convolutional codes using soft-decision signals, an area which is currently receiving more attention.

The analysis algorithm proposed in Reference [10] utilizes soft bit information and investigates its efficacy on standard convolutional codes. However, the coding rates explored in the paper are somewhat limited, making it challenging to handle complex real-world environments. Reference [11] presents a blind-recognition algorithm that utilizes soft information from the received sequence to estimate the posterior probability of syndromes. However, this approach encounters challenges in establishing a robust detection threshold.

With the rapid expansion in the amount of real-time data and the growing necessity for immediate decision-making, there has been an increased demand for efficient management and access to diverse, heterogeneous data [12]. The advent of deep learning has provided new opportunities to address this demand, enabling the automation and optimization of various tasks in signal processing. For instance, it has been used to design adaptive filters [13] and automatic noise reduction [14], enhancing the robustness of communication systems. In channel coding, deep learning can perform channel code recognition by leveraging the information obtained from the demodulation output. Various deep neural networks, such as TextCNN [15,16], have been proposed for channel code recognition. The types of recognition involved, however, are significantly limited, and the accuracy is relatively low, particularly under low-SNR conditions. Additionally, a deep learning method based on deep residual networks [17] has been proposed for blind recognition of convolutional code parameters from a given soft-decision sequence. However, this method fails to consider the possibility that the received encoded sequence may not start from the beginning of a complete codeword, which could impact the model’s recognition performance, especially in the presence of channel noise or other disturbances. To mitigate the issue of low accuracy associated with single-type neural networks, a novel channel recognition algorithm based on bi-directional long short-term memory (BiLSTM) and convolutional neural networks (CNN) has been proposed [7]. However, it solely discriminates among specific types of convolutional codes, LDPC codes, and polar codes. Table 1 summarizes the strengths and weaknesses of the various methods.

To enhance the recognition performance of channel coding, we implement an improved model architecture. The main contributions of this paper are as follows:
The ConvLSTM-TFN model is proposed, which seamlessly integrates convolutional layers [18], LSTM networks [19], and self-attention mechanisms [20]. By synergistically leveraging the strengths of these deep learning modules, our model exhibits enhanced capability in capturing channel-coding features, making it applicable to a wider range of coding types.We investigate the influence of input length, decision type (soft/hard), and constraint length on the accuracy of channel-coding recognition, utilizing convolutional codes as a case study. The advantages of soft-decision decoding and the broad applicability of this study within the field of coding are clarified.A dataset is generated encompassing 17 distinct convolutional code parameters that include both soft- and hard-decision sequences, along with randomly assigned encoding starting positions. This approach offers a potential avenue for the development of channel-coding datasets.


The remaining sections of this paper are organized as follows. Section 2 introduces the structure of a deep learning-based convolutional code blind-recognition system, along with an explanation of the basic theory of deep learning. Section 3 describes the dataset used for the experiments and explores the application of a deep neural network for blind recognition of convolutional codes. In Section 4, we analyze how different parameters affect the performance of our model in blind recognition on convolutional codes. Finally, Section 5 concludes this paper and discusses future works.

## 2. System Model

In this study, we propose a signal transmission system specifically designed for the identification of channel-coding parameters. Figure 1 illustrates the basic framework of a wireless communication system, wherein data undergo encoding, modulation, and transmission within the transmitting system. Subsequently, the dataset traverses a noisy wireless channel before being received at the other end, where it is demodulated and decoded. The channel code recognition module, highlighted in purple, constitutes a pivotal element within this system, as it assumes a central role in the identification of channel code types under non-cooperative communication settings.

In the following sections, we will explain the details of the transmitting and receiving systems, including the encoding scheme (convolutional coding) and modulation (BPSK), and the specific channel conditions assumed in this study (AWGN).

The message to be transmitted, ***M*** = (*m*_1_, *m*_2_, …, *m_k_*), is initially encoded into codewords ***c*** = (*c*_1_, *c*_2_, …, *c_k_*) with *n* bits (*k* < *n*), using a convolutional coding scheme. The codeword *c* is formed based on the generator matrix *G* as follows:(1)$$c_{1\times n}=M_{1\times k}\cdot G_{k\times n}$$

Convolutional codes can be represented by a generator matrix. For a convolutional code *C*(*n*, *k*, *l*), the generator matrix typically takes the following form:(2)$$G(D)=\left[\begin{array}{cccc} g_{11}(D) & g_{12}(D) & \ldots & g_{1k}(D)\\ g_{21}(D) & g_{22}(D) & \ldots & g_{2k}(D)\\ \vdots & \vdots & \ddots & \vdots \\ g_{n1}(D) & g_{n2}(D) & \ldots & g_{nk}(D) \end{array}\right]$$
where *D* represents a delay operator (commonly denoted as a shift register), and the generator polynomials, which consist of shift registers and modulo-2 addition.

Figure 2 illustrates the structure of a convolutional code. The diagram depicts the register structure of the convolutional encoder, showing how the input bits are processed through a series of shift registers and modulo-2 adders to generate the output codeword.

In the figure, a convolutional encoder is depicted with *L* levels of information bits, where *k* input bits are processed at each time step, resulting in a total of *k*L* bits. At each step, *n* output bits are produced. It is evident that the output of these *n* bits depends not only on the current *k* input bits but also on the preceding *L*-1 groups of input information bits. Hence, convolutional codes exhibit memory characteristics with a memory depth of *L*.

After encoding, the modulation scheme employed is binary phase shift keying (BPSK), while the communication channel exhibits addictive white Gaussian noise (AWGN). The presence of AWGN introduces distortions to the received signal, thereby augmenting the probability of errors and posing challenges in accurately decoding the transmitted data.

At the receiver, due to the imperfections in the channel, transmitter, and receiver sensors, the received signal *s*′ is disturbed, resulting in the difference between the decoded information ***M***′ and the original information ***M***. Channel coding is one of the key methods used to mitigate these distortions. In low-SNR environments, achieving reliable demodulation often requires manually setting thresholds for optimal decision-making, which is a challenging task. Therefore, by utilizing soft decision $\hat{c}$ rather than the hard decision results ***c***′, we can automate the threshold adjustment, effectively improving the system’s robustness and enhancing its recognition accuracy. Simultaneously, we consider the situation in which the received encoded sequence is incomplete, a which implies that it may not start at the actual starting position of the encoded sequence. This adds complexity to the recognition of encoding types.

In this paper, we propose a method based on the ConvLSTM-TFN, which effectively addresses the task of identifying channel code parameters under low-SNR conditions. Our study specifically focuses on convolutional codes as the predefined channel-coding scheme and conducts parameter identification under an AWGN channel with BPSK modulation. Importantly, our proposed method is versatile and can be applied to identify parameters under different channels, modulation schemes, and coding methods. While we have chosen conditions specific to a single-channel code parameter identification task, our approach exhibits generalizability and potential for broader applications.

## 3. Proposed Method

### 3.1. ConvLSTM-TFN

Convolutional codes exhibit long-term dependencies, in contrast to other channel codes, such as linear block codes, which only exhibit block-by-block dependencies [21]. This dependency is manifested during the encoding process, in which each output codeword is influenced not only by the current input symbol but also by the preceding input symbols. Therefore, to enhance the effectiveness of the blind recognition of convolutional code parameters, a deep learning network that is capable of handling long-term dependencies in sequences, such as long short-term memory (LSTM), is necessitated. Moreover, convolutional codes showcase local correlation and structure through their ability to convert input bits into multiple output bits at specific coding rates. Consequently, we have incorporated convolutional layers for feature extraction, enabling the effective capture of these short-term features for a more accurate recognition of the patterns and structures of convolutional codes. Simultaneously, we have added a self-attention mechanism to enhance the overall recognition performance of the model.

The proposed model is a hybrid architecture that integrates convolutional layers, an LSTM module, and a self-attention mechanism specifically designed for sequence classification tasks. The network processes input sequences of size 256 × 1; the details of each layer are shown in Table 2. Initially, two 1D convolutional layers with ReLU [22] activation are applied to extract local features from the input data. These extracted features are then passed to a multi-layer LSTM module with a hidden size of 768 to effectively capture long-term dependencies. Subsequently, a self-attention mechanism assigns different weights to the hidden states, enabling the model to focus on the most relevant parts of the sequence. To ensure robustness and generalization capability, dropout and batch normalization techniques are applied before the resulting context vector is fed into a fully connected layer for accurate classification.

The network uses the ReLU activation function in all layers except the output, where a softmax activation is applied to predict blind-recognition class probabilities. Cross-entropy loss [23] is used as the objective function to optimize the model, while an Adam optimizer [24] with learning rate scheduling ensures stable training. Regularization techniques, including dropout and batch normalization, are employed to improve generalization and stabilize learning.

In summary, our proposed ConvLSTM-TFN integrates LSTM, CNN, and self-attention mechanisms, leveraging the advantages of CNNs in local feature extraction, the advantages of LSTMs in capturing long-term dependencies in data sequences, and the advantages of self-attention mechanisms in establishing global dependencies. This synergy expands the receptive field, with the objective of improving performance in channel-coding blind recognition.

Figure 3 shows the overall network framework of the convolutional code blind recognition, illustrating the key modules and data flow between them.

In the subsequent sections, we will provide a comprehensive exposition of each component within the ConvLSTM-TFN, commencing with the LSTM component for sequence processing, followed by the convolutional layers for local feature extraction, and concluding with the self-attention mechanism for improved feature integration.

#### 3.1.1. Feature Extraction Layers

In this work, we use convolutional layers as a feature-extraction method for convolutional codes. Through the convolution operation and parameter sharing, convolutional layers can efficiently extract local features of convolutional codes. The local features inherent in convolutional codes can be observed in Figure 2, where the current output of the convolutional code is closely related to ***G***(*D*), although it is also influenced by the values stored in the registers (i.e., previous inputs). Convolutional layers have strong feature-extraction capabilities and do not require complex data preprocessing, making them particularly well-suited for one-dimensional signal recognition. Therefore, we employ two convolutional layers to capture higher-level abstract features. The convolution operation of the neurons can be mathematically expressed as follows:(3)$$y[i]=\sum_{j=0}^{k-1}w[j]\cdot x[i+j]+b$$
where *y*[*i*] represents the output, *w*[*i*] denotes the convolutional kernel weights, *x*[*i + j*] signifies the input data, and *b* is the bias term.

#### 3.1.2. LSTM Layers

Convolutional codes exhibit strong temporal dependencies and sequential structures, which makes the use of LSTM networks a suitable choice for their processing and training. LSTMs can capture long-term dependencies in sequential data, effectively handling the complex structures and patterns found in convolutional code sequences. Compared to standard RNNs, LSTMs introduce mechanisms such as input gates, forget gates, and output gates, as well as a cell state, allowing them to better manage long-range dependencies in sequences.

The forget gate controls which information should be discarded from the cell state. It takes the current input ***X****_t_* and the previous hidden state ***h****_t_* as inputs, and after passing through an activation function, it produces an output vector ***f****_t_*. The elements of ***f****_t_* close to zero indicate the information to be forgotten, while those closer to one represent the information that should be retained.(4)$$f_{t}=\sigma (W_{f}\cdot [h_{t-1},X_{t}]+b_{f})$$
where σ represents the sigmoid activation function. ***W****_f_* and ***b****_f_* are weight and bias.

The input gate determines which information is written into the cell state. The new cell state is formed by combining the information to be remembered with the previous cell memory.(5)$$\begin{array}{l} i_{t}=\sigma (W_{i}\cdot [h_{t-1},X_{t}]+b_{i})\\ {\tilde{C}}_{t}=\tanh(W_{c}\cdot [h_{t-1},X_{t}]+b_{c}) \end{array}$$
where ***i****_t_* controls how much new information is written, while $\tilde{C}$*_t_* represents the current input information.

The cell state is the core of the LSTM, which carries long-term memory information, with the interactions represented by element-wise multiplication.(6)$$C_{t}=f_{t}\times C_{t-1}+i_{t}\times {\tilde{C}}_{t}$$

The output gate regulates the information output from the cell state, and the resulting hidden state ***h****_t_* is utilized in the computations for the next time step.(7)$$\begin{array}{l} o_{t}=\sigma (W_{o}\cdot [h_{t-1},X_{t}]+b_{o})\\ h_{t}=o_{t}\times \tanh(C_{t}) \end{array}$$

#### 3.1.3. Self-Attention Mechanism

Due to the independence of the characteristics of convolutional codes from the distance between sequence elements, the network incorporates a self-attention mechanism to enhance the model’s ability to capture global information by expanding the receptive field. In the self-attention mechanism, each element in the input sequence interacts with all other elements, enabling the model to capture the relationships between them regardless of their positions in the sequence. Additionally, the self-attention mechanism allows the model to weigh the importance of different elements in the input sequence when making predictions.

Mathematically, the self-attention process can be described as follows:

Input Representation: Given an input sequence ***X***∈R^*n*×*d*^, where *n* is the sequence length and *d* is the dimensionality of each input vector.

Calculating Attention Scores: The attention scores are computed using three learned linear transformations: the Query (***Q***), Key (***K***), and Value (***V***):(8)$$\begin{array}{ccc} Q=XW_{Q}, & K=XW_{K}, & V=XW_{V} \end{array}$$
here, ***W_Q_***, ***W_K_***, and ***W_V_*** are weight matrices.

Attention Weights: The attention weights are obtained by applying the softmax function to the scaled dot product of the Query and Key matrices:(9)$$Attention(Q,K,V)=softmax(\frac{QK^{T}}{\sqrt{d_{k}}})V$$
where ***d****_k_* is the dimensionality of the Key vectors, which helps to stabilize gradients during training.

Output: The output of the self-attention layer is a weighted sum of the Value vectors, allowing the model to attend to the most relevant parts of the input sequence.

### 3.2. Data Processing

This paper considers 17 types of convolutional codes, with code rates of 1/2, 1/3, and 1/4, along with the different constraint defined by our dataset. Specifically, we use convolutional codes with a rate of 1/2 and constraint lengths ranging from 3 to 9, rate 1/3 convolutional codes with constraint lengths ranging from 2 to 6, and rate 1/4 convolutional codes with constraint lengths ranging from 2 to 6. Table 3 presents the octal representation of the generating polynomials for these different codes.

In this work, the dataset required for the simulation is generated using MATLAB 2020a. The data generation process consists of three main steps. First, random signals are encoded using the convolutional code parameters shown in Table 3. Second, the encoded signals are modulated using BPSK modulation and transmitted through an AWGN channel. Finally, the received channel signals are demodulated using a BPSK demodulator, and the soft information bits from the demodulation process form the dataset. When the SNR level ranges from −20 dB to 20 dB, for each 2 dB increase in SNR, 5000 samples are generated for each convolutional code parameter, with 80% of the samples being used for training and the remainder being used for validation. Consequently, our dataset comprises a total of 1,785,000 training samples and 1,428,000 testing samples; each sample has a length of 256.

The process of sample generation is as follows:Generate a binary random sequence with a length of 300 bits.Perform the corresponding convolutional encoding based on the parameters in Table 3.Modulate the encoded sequence using BPSK modulation to generate the modulated signal.Add AWGN to generate the noisy signal, simulating the effects of channel transmission.Demodulate the signal to obtain the soft-decision results corrupted by interference.Randomly select a starting point from the first 0 to 20 numbers in the sequence and extract 256 bits as a sample sequence.Repeat steps 1–6 to generate the complete dataset.

## 4. Experiment Evaluation

### 4.1. Experimental Environment and Evaluation Metrics

In this paper, the proposed model is developed and trained using PyTorch 2.0.1. The specific hyperparameters of the deep learning model are detailed in Table 2. The model is trained and tested on a computer equipped with an AMD Ryzen 9 5950X 16-core processor, 32 threads, 32 GB of RAM, and an NVIDIA GeForce RTX 3090 Ti GPU.

For each experiment, we perform five runs and calculate the average of the results. The evaluation metrics include OA (overall accuracy), precision, and the F1 score, which provide a comprehensive assessment of the model’s performance across different aspects. These metrics are widely used for performance evaluation and can be computed using standard formulas, as outlined in [25]. Since our dataset is evenly distributed, OA is equivalent to the recall value. Therefore, we do not include recall as a separate evaluation metric.

### 4.2. Impact of Data Characteristics

#### 4.2.1. Impact of Sample Length on Performance

The length of input samples plays a critical role in the blind recognition of convolutional codes, as increasing the sequence length significantly increases both computational complexity and training time. To analyze the impact of sample length on model performance, we extended the dataset described in Section 3.2 by incorporating samples with lengths ranging from 64 to 512 bits and trained them using the ConvLSTM-TFN model.

The experimental results in Table 4 demonstrate that the model’s recognition accuracy improves as sample lengths increase. However, beyond a certain threshold, the rate of accuracy improvement diminishes noticeably. As shown in Figure 4, in the low-SNR range (SNR < 0 dB), longer sequences offer limited advantages, with only marginal improvements found in accuracy compared to shorter sequences. Specifically, 64-bit and 128-bit sequences perform poorly in this region, while the difference in performance between 256-bit and 512-bit sequences is negligible. These findings suggest that, under extreme noise conditions, further increasing sequence length has limited effectiveness in enhancing model performance.

In the high-SNR range (SNR > 10 dB), classification accuracy for all sequence lengths converges to near-perfect levels, ultimately approaching 100%. This suggests that, in high-SNR conditions, signal information is sufficient for accurate classification, rendering sequence length variations nearly irrelevant to performance. Notably, the longer sequences of 512 bits exhibit slightly lower performance than shorter sequences in high-SNR conditions, likely due to increased model complexity and redundant information.

Overall, the results indicate that extending sequence length does not consistently result in significant improvements in classification accuracy, especially in low- and moderate-SNR ranges. Although longer sequences provide additional information, their effectiveness may be limited by the presence of noise and the model’s processing capability. Moreover, longer sequences considerably augment computational complexity and training time, imposing greater demands on model design and resource consumption. Considering the trade-off between classification performance and computational cost, we select 256-bit sequences as the primary training samples to strike a balance between efficiency and effectiveness.

#### 4.2.2. Pre-Class Performance

Figure 5 illustrates the variation in recognition accuracy for each type of convolutional code versus SNR under the ConvLSTM-TFN model.

The recognition accuracy for all convolutional code types approaches nearly 100% when the SNR is greater than or equal to 6 dB, indicating the remarkable efficacy of the proposed method in identifying convolutional codes under typical channel conditions.

Additionally, the performance of the classification model is significantly influenced by the constraint length *l* of convolutional codes. Under low-SNR conditions, shorter constraint lengths (e.g., *l* = 2, *l* = 3) demonstrate greater robustness. These shorter codes typically exhibit a faster improvement in accuracy, with performance “breakpoints” occurring at lower SNR values (e.g., around −2 dB). In contrast, convolutional codes with longer constraint lengths (e.g., *C*(2, 1, 9)) tend to perform less effectively in low-SNR environments but maintain excellent classification accuracy under high-SNR conditions.

The findings suggest that the complexity of convolutional codes exhibits different strengths across varying SNR levels. This observation suggests that models must strike a balance between robustness and complexity to adapt to dynamic noise environments. Notably, our proposed ConvLSTM-TFN network effectively captures and leverages the intricate relationships between signal features, enabling reliable classification, even under challenging conditions. By combining feature extraction, sequential modeling, and self-attention mechanisms, our network demonstrates exceptional adaptability across diverse SNR levels. Shorter constraint lengths benefit from the network’s robustness in noisy conditions, while longer constraint lengths leverage the network’s capacity to extract detailed features in high-quality channels. These findings underscore the capability of our method to generalize effectively while maintaining high classification performance, proving its suitability for real-world signal processing tasks.

### 4.3. Impacts of LSTM Depth and Hidden Size on Performance

The number of LSTM layers and the hidden size have significant impacts on the system’s blind-recognition performance. We analyzed the effects of different hidden sizes with a four-layer LSTM architecture, as well as the impact of a 256 hidden size across multiple LSTM layers.

The results presented in Table 5 indicate that increasing the hidden size generally enhances model performance; however, this improvement exhibits diminishing returns beyond a certain threshold. When LSTM layers are fixed at four, the model achieves its highest OA, at 65.57%, and F1 score, at 65.84%, with a hidden size of 1024. In contrast, a hidden size of 768 provides nearly identical results (65.03% OA and 65.86% F1) while incurring lower computational cost. This finding suggests that a hidden size of 768 strikes an effective balance between performance and efficiency.

In comparing various configurations of LSTM layers, we observe that an increase in the number of layers does not necessarily correlate with improved overall performance. Specifically, when utilizing two layers, the model performs significantly worse, achieving only 62.79% OA and a 63.81% F1 score. This indicates that deeper architectures may be advantageous for this particular task. Conversely, expanding the layers to six (with a hidden size of 768) yields the highest precision, at 67.01%, yet results in a decrease in OA to 63.46%, which suggests potential overfitting issues. In contrast, employing four LSTM layers strikes the best overall balance, consistently delivering robust performance across all evaluated metrics.

Based on preliminary observations, a hidden size of 768 achieves the optimal balance between performance and computational efficiency. Current experimental trends indicate that larger hidden sizes yield diminishing returns, rendering it unnecessary to exhaustively test all potential configurations.

From these results, it is evident that while a hidden size of 1024 achieves slightly superior outcomes, the difference is minimal when compared to a hidden size of 768. The latter represents a more practical choice due to its reduced computational complexity. Additionally, increasing the number of LSTM layers beyond four does not enhance either the OA or F1 score and may even lead to overfitting. Therefore, we conclude that an optimal configuration consists of four LSTM layers and a hidden size of 768, effectively balancing accuracy, precision, and efficiency.

### 4.4. Performance Comparison Across Models

In our experiments, we compared the proposed ConvLSTM-TFN against DRN [17], CCR-Net [26], 1-D CNN, and TextCNN. Among these networks, DRN and CCR-Net are specifically designed for channel-coding recognition tasks, demonstrating exceptional performance in this domain and serving as representative works among the existing methods. In contrast, 1-D CNN and TextCNN are general-purpose networks, each with a wide range of applications. A comparison with these networks effectively highlights the advanced capabilities of our proposed network in the realm of channel-coding blind recognition. The dataset used for evaluation is the one generated in Section 3.2, along with a hard-decision counterpart generated under identical conditions. Each sample in the dataset contains a noisy version of the input signal as well as the corresponding coding information.

Figure 6a depicts the accuracy curves of each network under different decision schemes and varying SNR conditions. When the SNR > −2 dB, our network significantly outperforms all other models, demonstrating its robustness and efficiency. Although CCR-Net exhibits slightly superior performance, compared to our network, in extremely low-SNR conditions, resulting in an OA 1.86% higher than ConvLSTM-TFN, it fails to maintain this advantage across the entire SNR range. This is also evident in the median line of the box plot presented in Figure 6b, where ConvLSTM-TFN demonstrates the best performance in convolutional code blind recognition. Statistical analysis indicates that ConvLSTM-TFN achieves an accuracy exceeding 90% under practical channel conditions, with an SNR above 0 dB. Within the SNR range of 0–20 dB, the average blind-recognition accuracy of ConvLSTM-TFN reaches an impressive 98.7%. In contrast, our network excels under higher-SNR conditions, significantly outperforming other models and highlighting its superior capability in handling complex environments.

Table 6 compares the accuracy across all networks under both soft-decision and hard-decision scenarios. Under soft-decision conditions, channel-coding recognition demonstrates overall superior performance compared to hard-decision conditions, with an approximate increase of 2% in OA for soft decision. Soft decision provides richer signal details, enabling complex networks to extract more comprehensive features and achieve higher classification accuracy. However, for simpler network architectures, such as TextCNN, the limited feature-extraction capability makes hard decision more advantageous for structured feature extraction, leading to higher OA in certain cases. Additionally, for basic networks like 1-D CNN, the decline in recognition performance is relatively modest as the SNR decreases, suggesting that lower-complexity networks can only partially leverage the benefits of soft-decision information.

In summary, the experiment’s results validate the advantages of soft decision in channel-coding recognition while highlighting the trade-off between network complexity and the type of input features.

Figure 7 illustrates the confusion matrices of each network under conditions where SNR ≥ 0 dB. It can be observed that although all networks exhibit high accuracy for certain coding types, there still remains a degree of confusion between different coding types. This confusion is particularly evident in some networks, where misclassifications occur more frequently among two or more coding types, significantly affecting the reliability of the decisions. However, in the ConvLSTM-TFN model, this confusion is markedly reduced, further validating the superior performance and robustness of our network in the coding classification task.

Moreover, considering that SNR ≥ 0 dB typically corresponds to the prevailing transmission conditions in practical communication systems, our network demonstrates reliable decision-making in this environment, thereby minimizing misclassifications and showcasing enhanced reliability and applicability. These results underscore the proficiency of our network in accurately classifying diverse coding types while effectively mitigating confusion, rendering it highly suitable for real-world applications.

In summary, the experimental results demonstrate the superior performance and robustness of the ConvLSTM-TFN network in channel-coding classification tasks. By effectively mitigating confusion between coding types, particularly under typical communication conditions (SNR ≥ 0 dB), our model achieves enhanced reliability and classification accuracy compared to other networks. This emphasizes the practical applicability of our approach and its potential for deployment in real-world communication systems. Additionally, the results underscore the significance of balancing network complexity and robustness to optimize classification performance across diverse conditions.

### 4.5. Ablation Study

Table 7 presents a comparison of performance between the baseline model (which comprises only the backbone network) and the ConvLSTM-TFN model, which incorporates a self-attention mechanism. The results indicate that the integration of self-attention consistently enhances all evaluation metrics.

Specifically, the OA shows an increase from 64.28% to 65.03%, indicating a modest yet meaningful improvement in overall classification performance. Notably, the most substantial improvements are observed in precision and F1 score. The precision value rises from 61.32% to 66.73%, indicating that the model utilizing self-attention generates fewer false positive predictions and achieves higher reliability in its positive classifications. Similarly, the F1 score improves from 62.77% to 65.86%, reflecting a more favorable balance between precision and recall in the enhanced model.

These results indicate that the self-attention mechanism primarily enhances the model’s capacity to distinguish complex backgrounds and easily confused categories. The introduction of self-attention effectively bolsters feature extraction and sequence modeling, leading to notable improvements in precision and model robustness. Although the overall accuracy improvement is relatively modest, there has been a significant increase in both the model’s precision and its overall performance. Consequently, when compared to the baseline model, the ConvLSTM-TFN integrated with the self-attention mechanism exhibits superior performance and represents a more effective configuration for modeling tasks.

## 5. Conclusions

In this study, we propose a novel channel-coding recognition network, referred to as ConvLSTM-TFN. This method integrates convolutional layers, LSTM networks, and a self-attention mechanism to effectively capture and distinguish the characteristics of various channel-coding types. The hybrid model addresses limitations associated with single-component approaches and demonstrates superior performance by utilizing soft-decision sequences. The model demonstrates a blind-recognition accuracy for convolutional codes exceeding 90% when the SNR is greater than 0 dB. Furthermore, it achieves an average accuracy of 98.7% within the SNR range of 0 to 20 dB. Experimental results demonstrate the superiority of our network compared to existing models, while also indicating significant reductions in inter-type confusion.

In future research, our primary object will be to prioritize the enhancement of performance under low-SNR conditions and expand the model’s capability to handle a wider variety of coding types, ensuring clear differentiation across diverse categories. We plan to further innovate in order to improve accuracy in noisy environments and validate the adaptability of the ConvLSTM-TFN model across an expanded range of channel-coding types.

## Figures and Tables

**Figure 1 sensors-25-01000-f001:**
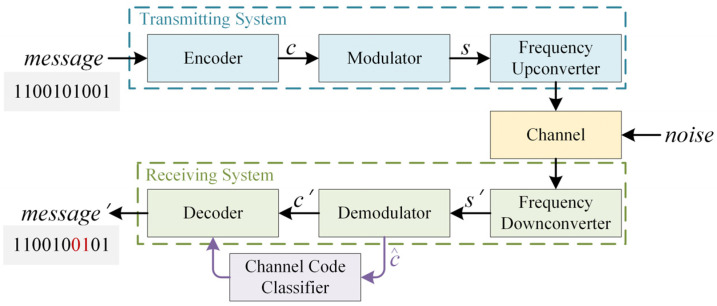
The structure of a wireless communication system.

**Figure 2 sensors-25-01000-f002:**
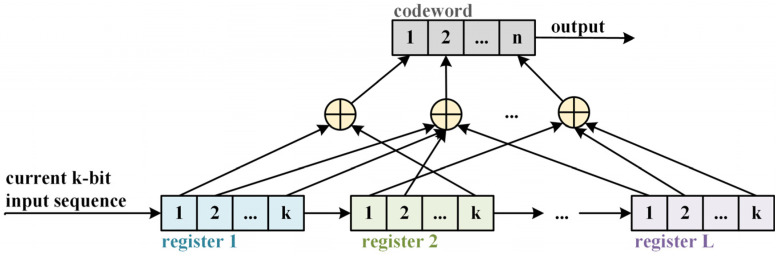
The register structure of a convolutional encoder.

**Figure 3 sensors-25-01000-f003:**
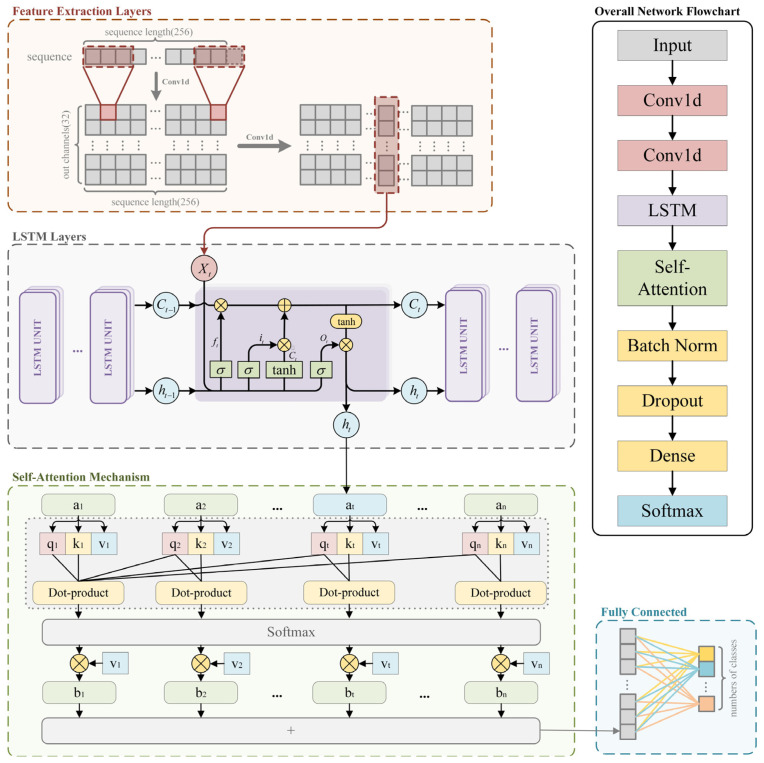
The structure of the ConvLSTM-TFN. The left side shows the detailed architecture of the network, while the right side presents the network flowchart. The colors in the flowchart correspond to different components of the detailed architecture, with the yellow section indicating a part that is not explicitly depicted in the detailed architecture on the left.

**Figure 4 sensors-25-01000-f004:**
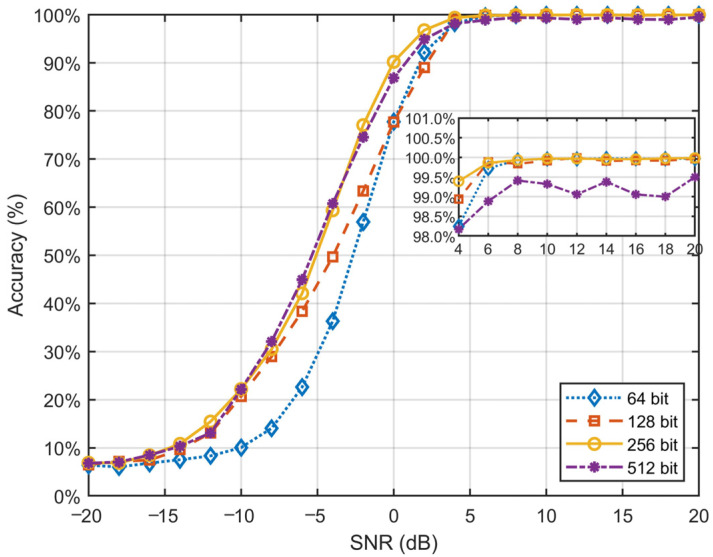
Performance versus example length in the samples.

**Figure 5 sensors-25-01000-f005:**
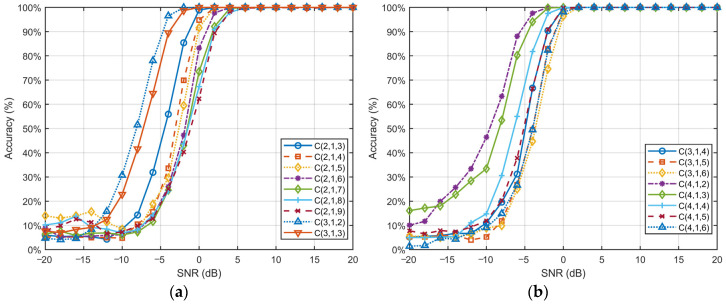
The recognition accuracy of each convolutional code type versus SNR in the ConvLSTM-TFN network. The coding types are presented in two separate sections, (**a**,**b**), which facilitates a more convenient comparison and analysis.

**Figure 6 sensors-25-01000-f006:**
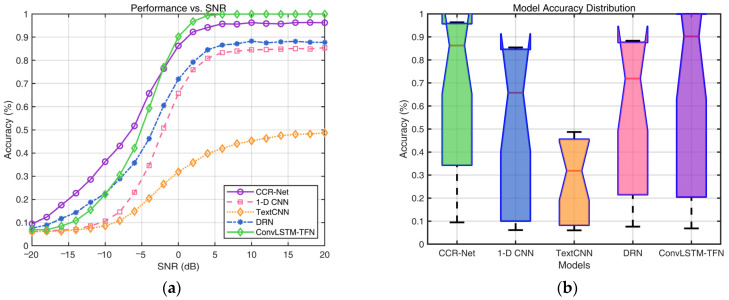
Performance comparison for various deep learning models in channel code recognition: (**a**) illustrates the recognition accuracy comparison among different methods versus SNR; (**b**) presents the box plots for each model.

**Figure 7 sensors-25-01000-f007:**
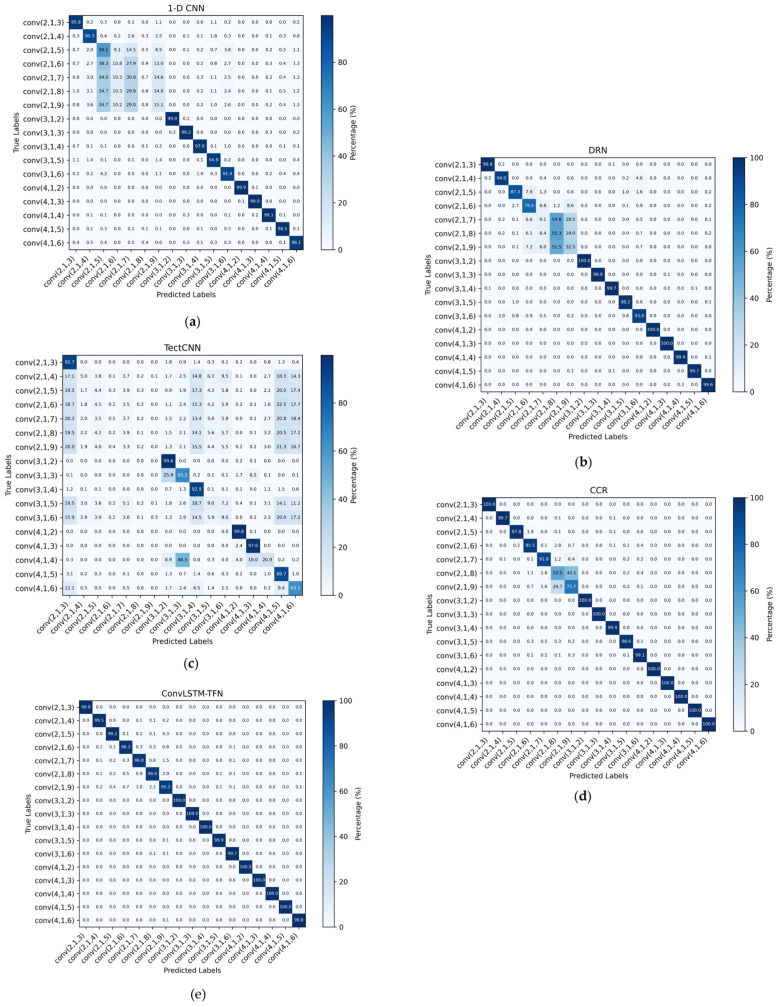
The confusion matrices for different methods, with SNR ≥ 0 dB: (**a**) 1-D CNN; (**b**) DRN; (**c**) TextCNN; (**d**) CCR; and (**e**) ConvLSTM-TFN.

**Table 1 sensors-25-01000-t001:** Comparison of methods: Code Types, Prior Knowledge, and Noise Resilience.

Method	Applicable Code Types	Prior Knowledge	Noise Resilience
Puncturing Patterns Detection [8]	Broad	Not Required	Low
Dual Code Principles [9]	Broad	Not Required	Low
Generalized Likelihood Ratio Test (GLRT) [10]	Limited to 1/2 rate convolutional codes	Need initial position of the codeword	Low
Syndrome Posterior Probability (SPP) [11]	Limited to 1/2 rate convolutional codes	Need initial position of the codeword	High
TextCNN [15]	Limited to three convolutional code types	Not Required	High
Deep Residual Network (DRN) [17]	Broad	Need initial position of the codeword	High
BiLSTM-CNN [7]	Limited to three code types	Not Required	High
ConvLSTM-TFN	Broad	Not Required	High

**Table 2 sensors-25-01000-t002:** Network configuration and parameter summary.

Layer	Type	Filter Size /Stride	Output Shape	Activation Function
Input	Input	/	(256, 1)	/
Conv1	Conv1d	3 × 1/1	(256, 32)	ReLU
Conv2	Conv1d	3 × 1/1	(256, 32)	ReLU
LSTM	LSTM (4 Layers)	/	(256, 768)	ReLU
Attention Encoder	Linear	/	(256, 32)	Tanh
Attention Decoder	Linear	/	(256, 1)	Softmax
Weighted Sum	Attention Output	/	(768)	/
Batch Norm	BatchNorm 1D	/	(768)	/
Dropout	Dropout (*p* = 0.5)	/	(768)	/
fc	Dense + Softmax	/	(17)	Softmax

**Table 3 sensors-25-01000-t003:** Detailed parameters of convolutional codes (Rate-1/2, Rate-1/3, and Rate-1/4).

Code Type	Code Rate	Constraint	Generators ^1^
C(2, 1, 3)	1/2	3	(5, 7)
C(2, 1, 4)		4	(13, 17)
C(2, 1, 5)		5	(27, 31)
C(2, 1, 6)		6	(53, 75)
C(2, 1, 7)		7	(133, 171)
C(2, 1, 8)		8	(247, 371)
C(2, 1, 9)		9	(561, 753)
C(3, 1, 2)	1/3	2	(1, 3, 3)
C(3, 1, 3)		3	(5, 7, 7)
C(3, 1, 4)		4	(13, 15, 17)
C(3, 1, 5)		5	(25, 33, 37)
C(3, 1, 6)		6	(47, 53, 75)
C(4, 1, 2)	1/4	2	(1, 1, 3, 3)
C(4, 1, 3)		3	(5, 5, 7, 7)
C(4, 1, 4)		4	(13, 13, 15, 17)
C(4, 1, 5)		5	(25, 27, 33, 37)
C(4, 1, 6)		6	(45, 53, 67, 77)

^1^ The generators use octal representation.

**Table 4 sensors-25-01000-t004:** Overall accuracy for each input sequence length.

Sequence Length	64-bit	128-bit	256-bit	512-bit
OA	59.17%	62.38%	65.03%	65.04%

**Table 5 sensors-25-01000-t005:** Impact of configuration choices on performance.

LSTM Layers	Hidden Size	OA	Precision	F1 Score
2	768	62.79%	64.86%	63.81%
4	256	61.67%	66.85%	63.01%
4	512	63.23%	65.97%	64.57%
4	768	65.03%	66.73%	65.86%
4	1024	65.57%	66.12%	65.84%
6	768	63.46%	67.01%	65.18%

**Table 6 sensors-25-01000-t006:** Recognition accuracy for different methods under different judgment conditions.

Network	OA (Soft-Decision)	OA (Hard-Decision)
CCR-Net	66.88%	63.74%
TextCNN	28.17%	34.90%
1-D CNN	50.85%	49.30%
DRN	56.78%	54.52%
ConvLSTM-TFN	65.03%	62.98%

**Table 7 sensors-25-01000-t007:** ConvLSTM-TFN ablation study.

Model	Backbone Network	Self- Attention Mechanism	OA	Precision	F1 Score
Baseline	√		64.28%	61.32%	62.77%
ConvLSTM-TFN	√	√	65.03%	66.73%	65.86%

## Data Availability

Data are contained within the article.

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
