# Peer review of "Blind Recognition of Convolutional Codes Based on the ConvLSTM Temporal Feature Network"

_sensors, 2025, doi:10.3390/s25041000_

Round 1
Reviewer 1 Report
Comments and Suggestions for Authors
This manuscript presents ConvLSTM-TFN, a hybrid deep learning model that combines convolutional layers, LSTM networks, and a self-attention mechanism for blind recognition of convolutional codes in wireless communication. However, several aspects require improvement to enhance the overall quality of the manuscript.
1. Highlight the research gaps to clearly identify the limitations of previous studies and demonstrate how the novel contributions of this research effectively address these gaps.
2. A comparative study table at the end of the introduction, summarizing the pros and cons of each method could strengthen the review.
3. The choice of integrating convolutional layers, LSTM, and a self-attention mechanism is reasonable, but the rationale for selecting specific configurations (e.g., four LSTM layers, hidden size 768) is not discussed. Justify these choices with references or experimental evidence.
4. The paper lacks justification for certain design choices, such as the selection of sequence length (256). A discussion of the trade-offs in choosing this length would be helpful.
5. Reporting metrics beyond accuracy, such as precision, recall, or F1-score, would give a more nuanced understanding of the model's behavior under ablated conditions.
6. The discussion on SNR levels is informative, but including visual comparisons (e.g., box plots) of model performance across methods would strengthen the analysis.
7. The confusion matrix in Figure 7 is unclear and difficult to interpret. Please provide a larger and more detailed version to ensure the data is easily readable and extractable.
8. If there is no comparative result analysis with prior works, it becomes difficult to assess the true impact of this study. Including a detailed comparative analysis with existing methods is essential to highlight how this work improves upon or differs from prior approaches in terms of performance, accuracy, efficiency, or robustness.
9. The abstract states that the method achieves over 90% recognition accuracy across 17 convolutional code types, with an average accuracy of 98.7%, but I could not find any justification or supporting evidence for this claim in the results section. Please ensure the results section provides detailed data or analysis to validate this statement.
Author Response
|
Comments 1: Highlight the research gaps to clearly identify the limitations of previous studies and demonstrate how the novel contributions of this research effectively address these gaps. |
|||||||||||||||||||||||||||||||||||||||||||||||||||||
|
Response 1: Thank you for your valuable feedback. We agree with this suggestion. To address this, we have highlighted the research gaps more explicitly in the revised manuscript, clearly identifying the limitations of previous studies. Additionally, we have emphasized how the novel contributions of this research effectively address these gaps. This change can be found in the revised manuscript on page 2, Sec. 1, paragraph 3-5 and 7, lines 46-63 and 85-99. ---Change in Manuscript--- Page 2, Sec. 1, paragraph 3-5, lines 46-63: Convolutional codes typically possess good error-correction capabilities, but error propagation may occur if the decoder incorrectly selects a wrong code or decoding path. The complexities highlighted here emphasize the significance of precise recognition of convolutional codes. Convolutional code recognition and parameter estimation methods have predominantly relied on the Galois field GF(2). Hard decision-based convolutional code recognition methods have been well-developed, enabling the recognition of various convolutional codes without necessitating prior knowledge [8-9][1]. Nevertheless, these approaches generally demonstrate limited resilience to noise. Due to the suboptimal performance of convolutional code recognition under low signal to noise ratio (SNR), there has been a shift towards parameter identification of convolutional codes using soft decision signals, which is currently receiving more attention. The analysis algorithm proposed in reference [10][2] utilizes soft bit information and investigates its efficacy on standard convolutional codes. However, the coding rates studied in the paper are somewhat limited, making it challenging to handle complex real-world environments. Reference [11][3] presents a blind recognition algorithm that utilizes soft information from the received sequence to estimate the posterior probability of syndromes. However, this approach encounters challenges in establishing a robust detection threshold. Page 2, Sec. 1, paragraph 7, lines 85-99: To enhance the recognition performance of channel coding, we implement an improved model architecture. The main contributions of this paper are as follows: 1. The ConvLSTM-TFN model is proposed, which seamlessly integrates convolutional layers [18][4], LSTM networks [19][5], and self-attention mechanisms [20][6]. By synergistically leveraging the strengths of these deep learning modules, our model exhibits enhanced capability in capturing channel coding features, making it applicable to a wider range of coding types. 2. We investigate the influence of input length, decision type (soft/hard), and constraint length on the accuracy of channel coding recognition, utilizing convolutional codes as a case study. The advantages of soft decision decoding and the broad applicability of this study within the field of coding are clarified. 3. A dataset is generated, encompassing 17 distinct convolutional code parameters that include both soft and hard decision sequences, along with randomly assigned encoding starting positions. This approach offers a potential avenue for the development of channel coding datasets.
|
|||||||||||||||||||||||||||||||||||||||||||||||||||||
|
Comments 2: A comparative study table at the end of the introduction, summarizing the pros and cons of each method could strengthen the review. |
|||||||||||||||||||||||||||||||||||||||||||||||||||||
|
Response 2: Thank you for the suggestion. We agree that a comparative research table can help clarify the limitations of previous studies and highlight the contributions of our work. In response, we have added a table comparing prior research and our study in terms of Applicable Code Types, Prior Knowledge, and Noise Resilience. This comparison aims to better contextualize our contributions and demonstrate the unique aspects of our approach. This change can be found in the revised manuscript on page 3, Sec. 1, Table 1, line 100. ---Change in Manuscript--- Table 1. Comparison of Methods: Code Types, Prior Knowledge, and Noise Resilience.
|
|||||||||||||||||||||||||||||||||||||||||||||||||||||
|
Comments 3: The choice of integrating convolutional layers, LSTM, and a self-attention mechanism is reasonable, but the rationale for selecting specific configurations (e.g., four LSTM layers, hidden size 768) is not discussed. Justify these choices with references or experimental evidence. |
|||||||||||||||||||||||||||||||||||||||||||||||||||||
|
Response 3: Agree. Proposing the considerations for model parameter selection can better demonstrate the rationality of the model structure. We have added this experiment, which is essential. This change can be found in the revised manuscript on page 11-12, Sec. 4.3, lines 376-408. ---Change in Manuscript--- 4.3. Impact of LSTM Depth and Hidden Size on Performance The number of LSTM layers and the hidden size have a significant impact on the system's blind recognition performance. We analyzed the effects of different hidden sizes with a 4-layer LSTM architecture, as well as the impact of a 256 hidden size across multiple LSTM layers. The results presented in Table 5 indicate that increasing the hidden size generally enhances model performance; however, this improvement exhibits diminishing returns beyond a certain threshold. When LSTM layers are fixed at four, the model achieves its highest OA of 65.57% and F1 score of 65.84% with a hidden size of 1024. In contrast, a hidden size of 768 provides nearly identical results (65.03% OA and 65.86% F1) while incurring lower computational cost. This finding suggests that a hidden size of 768 strikes an effective balance between performance and efficiency. Table 5. Impact of configurations choices on performance.
In comparing various configurations of LSTM layers, we observe that an increase in the number of layers does not necessarily correlate with improved overall performance. Specifically, when utilizing two layers, the model performs significantly worse, achieving only 62.79% OA and 63.81% F1 score. This indicates that deeper architectures may be advantageous for this particular task. Conversely, expanding the layers to six (with a hidden size of 768) yields the highest Precision at 67.01%, yet results in a decrease OA to 63.46%, which suggests potential overfitting issues. In contrast, employing four LSTM layers strikes the best overall balance, consistently delivering robust performance across all evaluated metrics. Based on preliminary observations, a hidden size of 768 achieves the optimal balance between performance and computational efficiency. Current experimental trends indicate that larger hidden sizes yield diminishing returns, rendering it unnecessary to exhaustively test all potential configurations. From these results, it is evident that while a hidden size of 1024 achieves slightly superior outcomes, the difference is minimal when compared to a hidden size of 768. The latter represents a more practical choice due to its reduced computational complexity. Additionally, increasing the number of LSTM layers beyond four does not enhance either the OA or F1 score and may even lead to overfitting. Therefore, we conclude that an optimal configuration consists of four LSTM layers and a hidden size of 768, effectively balancing accuracy, precision, and efficiency.
|
|||||||||||||||||||||||||||||||||||||||||||||||||||||
|
Comments 4: The paper lacks justification for certain design choices, such as the selection of sequence length (256). A discussion of the trade-offs in choosing this length would be helpful. |
|||||||||||||||||||||||||||||||||||||||||||||||||||||
|
Response 4: Thank you for your valuable feedback. Upon reviewing the manuscript, we realized that the experiment you mentioned was indeed included, but it was not clearly labeled in the original version. We apologize for any confusion caused by the mislabeling of the section title. The experiment is now clearly presented in the revised manuscript under section 4.2.1, "Impact of Sample Length on Performance," which explores the influence of sample length on recognition performance across 64-bit to 512-bit samples. I hope this clarifies the matter. This change can be found in the revised manuscript on page 9-10, Sec. 4.2.1, line 315-346. ---Change in Manuscript--- 4.2.1. Impact of Sample Length on Performance The length of input samples plays a critical role in the blind recognition of convolutional codes, as increasing the sequence length significantly raises both computational complexity and training time. To analyze the impact of sample length on model performance, we extended the dataset described in Section 3.2 by incorporating samples with lengths ranging from 64 to 512 bits and trained them using the ConvLSTM-TFN model. The experimental results demonstrate that the model’s recognition accuracy improves as sample lengths increase. However, beyond a certain threshold, the rate of accuracy improvement diminishes noticeably. In the low SNR range (SNR < 0 dB), longer sequences offer limited advantages, with only marginal improvements in accuracy compared to shorter sequences. Specifically, 64-bit and 128-bit sequences perform poorly in this region, while the difference in performance between 256-bit and 512-bit sequences is negligible. These findings suggest that, under extreme noise conditions, further increasing sequence length has limited effectiveness in enhancing model performance.
Figure 4. Performance versus example length in samples. In the high SNR range (SNR > 10 dB), classification accuracy for all sequence lengths converges to near-perfect levels, ultimately approaching 100%. This suggests that, in high SNR conditions, signal information is sufficient for accurate classification, rendering sequence length variations nearly irrelevant to performance. Notably, longer sequences of 512 bits exhibit slightly lower performance than shorter sequences in high SNR conditions, likely due to increased model complexity and redundant information. Table 4. Overall accuracy for each input sequence length.
Overall, the results indicate that extending sequence length does not consistently result in significant improvements in classification accuracy, especially in low and moderate SNR ranges. Although longer sequences provide additional information, their effectiveness may be limited by the presence of noise and the model's processing capability. Moreover, longer sequences considerably augment computational complexity and training time, imposing greater demands on model design and resource consumption. Considering the trade-off between classification performance and computational cost, we select 256-bit sequences as the primary training samples to strike a balance between efficiency and effectiveness.
|
|||||||||||||||||||||||||||||||||||||||||||||||||||||
|
Comments 5: Reporting metrics beyond accuracy, such as precision, recall, or F1-score, would give a more nuanced understanding of the model's behavior under ablated conditions. |
|||||||||||||||||||||||||||||||||||||||||||||||||||||
|
Response 5: Thank you for your insightful feedback. We agree that reporting additional metrics such as precision, recall, and F1-score would provide a more comprehensive understanding of the model's performance, especially under ablated conditions. In response to your comment, we have included precision, and F1-score in the revised manuscript to complement the accuracy metric. Since our dataset is evenly distributed, OA is equivalent to the recall value. Therefore, we do not include recall as a separate evaluation metric. These additional metrics offer a more nuanced evaluation of the model’s behavior and help in understanding its performance across different classes and under varying conditions. This change can be found in the revised manuscript on page 9, 11-12 and 14-15, Sec. 4.1, 4.3 and 4.5, lines 308-313, 376-408, 478-499. ---Change in Manuscript--- Page 9, Sec. 4.1, paragraph 2, lines 308-313 4.1. Experimental Environment and Evaluation Metrics In this paper, the proposed model is developed and trained using PyTorch. The specific hyperparameters of the deep learning model are detailed in Table 2. The model is trained and tested on a computer equipped with an AMD Ryzen 9 5950X 16-core processor, 32 threads, 32 GB of RAM, and an NVIDIA GeForce RTX 3090 Ti GPU. For each experiment, we perform five runs and calculate the average of the results. The evaluation metrics include OA (overall accuracy), precision, and the F1 score, which provide a comprehensive assessment of the model's performance across different aspects. These metrics are widely used for performance evaluation and can be computed using standard formulas as outlined in [25][14]. Since our dataset is evenly distributed, OA is equivalent to the recall value. Therefore, we do not include recall as a separate evaluation metric. Page 11-12, Sec. 4.3, lines 376-408 4.3. Impact of LSTM Depth and Hidden Size on Performance The number of LSTM layers and the hidden size have a significant impact on the system's blind recognition performance. We analyzed the effects of different hidden sizes with a 4-layer LSTM architecture, as well as the impact of a 256 hidden size across multiple LSTM layers. The results presented in Table 5 indicate that increasing the hidden size generally enhances model performance; however, this improvement exhibits diminishing returns beyond a certain threshold. When LSTM layers are fixed at four, the model achieves its highest OA of 65.57% and F1 score of 65.84% with a hidden size of 1024. In contrast, a hidden size of 768 provides nearly identical results (65.03% OA and 65.86% F1) while incurring lower computational cost. This finding suggests that a hidden size of 768 strikes an effective balance between performance and efficiency. Table 5. Impact of configurations choices on performance.
In comparing various configurations of LSTM layers, we observe that an increase in the number of layers does not necessarily correlate with improved overall performance. Specifically, when utilizing two layers, the model performs significantly worse, achieving only 62.79% OA and 63.81% F1 score. This indicates that deeper architectures may be advantageous for this particular task. Conversely, expanding the layers to six (with a hidden size of 768) yields the highest Precision at 67.01%, yet results in a decrease OA to 63.46%, which suggests potential overfitting issues. In contrast, employing four LSTM layers strikes the best overall balance, consistently delivering robust performance across all evaluated metrics. Based on preliminary observations, a hidden size of 768 achieves the optimal balance between performance and computational efficiency. Current experimental trends indicate that larger hidden sizes yield diminishing returns, rendering it unnecessary to exhaustively test all potential configurations. From these results, it is evident that while a hidden size of 1024 achieves slightly superior outcomes, the difference is minimal when compared to a hidden size of 768. The latter represents a more practical choice due to its reduced computational complexity. Additionally, increasing the number of LSTM layers beyond four does not enhance either the OA or F1 score and may even lead to overfitting. Therefore, we conclude that an optimal configuration consists of four LSTM layers and a hidden size of 768, effectively balancing accuracy, precision, and efficiency. Page 14-15, Sec. 4.5, lines 478-499 4.5. Ablation Study Table 7 presents a comparison of the performance between the Baseline model (which solely comprises the backbone network) and the ConvLSTM-TFN model that incorporates a Self-Attention Mechanism. The results indicate that the integration of self-attention consistently enhances all evaluation metrics. Table 7. ConvLSTM-TFN Ablation Study.
Specifically, the OA demonstrates an increase from 64.28% to 65.03%, indicating a modest yet meaningful improvement in overall classification performance. Notably, the most substantial improvements are observed in precision and F1 Score. The precision rises from 61.32% to 66.73%, indicating that the model utilizing self-attention generates fewer false positive predictions and achieves higher reliability in its positive classifications. Similarly, the F1 Score improves from 62.77% to 65.86%, reflecting a more favorable balance between precision and recall in the enhanced model. These results indicate that the Self-Attention mechanism primarily enhances the model's capacity to distinguish complex backgrounds and easily confused categories. The introduction of Self-Attention effectively bolsters feature extraction and sequence modeling, leading to notable improvements in precision and model robustness. Although the overall accuracy improvement is relatively modest, there has been a significant increase in both the model's precision and its overall performance. Consequently, when compared to the baseline model, the ConvLSTM-TFN integrated with the Self-Attention mechanism exhibits superior performance and represents a more effective configuration for modeling tasks.
|
|||||||||||||||||||||||||||||||||||||||||||||||||||||
|
Comments 6: The discussion on SNR levels is informative, but including visual comparisons (e.g., box plots) of model performance across methods would strengthen the analysis. |
|||||||||||||||||||||||||||||||||||||||||||||||||||||
|
Response 6: Thank you for your helpful comment. We agree that including visual comparisons would enhance the analysis. To address this, we have added box plots to the revised manuscript to visually compare the model's performance across different methods at various SNR levels. These visualizations are now included in section 4.3, “Performance Comparison Across Models”, which we believe will provide a clearer understanding of the performance differences. This change can be found in the revised manuscript on page 12-13, Sec. 4.4, lines 421-437. ---Change in Manuscript---
Figure 6. Performance comparison of various deep learning models in channel code recognition. (a) shows the recognition accuracy comparison of different methods versus SNR; (b) shows the box plots of each model. Figure 6 (a) depicts the accuracy curves of each network under different decision schemes and varying SNR conditions. When the SNR > -2 dB, our network significantly outperforms all other models, demonstrating its robustness and efficiency. Although CCR-Net exhibits slightly superior performance than our network in extremely low-SNR conditions, resulting in OA 1.86% higher than ConvLSTM-TFN, it fails to maintain this advantage across the entire SNR range. This is also evident in the median line of the box plot presented in Figure 6 (b), where ConvLSTM-TFN demonstrates the best performance in convolutional code blind recognition. Statistical analysis indicates that ConvLSTM-TFN achieves an accuracy exceeding 90% under practical channel conditions with an SNR above 0 dB. Within the SNR range of 0-20 dB, the average blind recognition accuracy of ConvLSTM-TFN reaches an impressive 98.7%. In contrast, our network excels under higher-SNR conditions, significantly outperforming other models and highlighting its superior capability in handling complex environments.
|
|||||||||||||||||||||||||||||||||||||||||||||||||||||
|
Comments 7: The confusion matrix in Figure 7 is unclear and difficult to interpret. Please provide a larger and more detailed version to ensure the data is easily readable and extractable. |
|||||||||||||||||||||||||||||||||||||||||||||||||||||
|
Response 7: We sincerely apologize for the confusion caused by the unclear confusion matrix in Figure 7. To address this, I have provided a larger and more detailed version of the matrix in the revised manuscript. The updated version ensures that the data is more easily readable and interpretable. I hope this improves the clarity of the figure. This change can be found in the revised manuscript on page 14, Sec. 4.4, lines 475-477. ---Change in Manuscript--- |
|||||||||||||||||||||||||||||||||||||||||||||||||||||
|
|
|||||||||||||||||||||||||||||||||||||||||||||||||||||
|
Figure 7. The confusion matrix of different methods when SNR ≥ 0 dB: (a) 1-D CNN; (b) DRN; (c) TextCNN; (d) CCR; (e) ConvLSTM-TFN.
|
|||||||||||||||||||||||||||||||||||||||||||||||||||||
|
Comments 8: If there is no comparative result analysis with prior works, it becomes difficult to assess the true impact of this study. Including a detailed comparative analysis with existing methods is essential to highlight how this work improves upon or differs from prior approaches in terms of performance, accuracy, efficiency, or robustness. |
|||||||||||||||||||||||||||||||||||||||||||||||||||||
|
Response 8: Thank you for your valuable comment. We agree that a detailed comparative analysis with prior works is essential to assess the true impact of this study. We apologize for not emphasizing this point more clearly earlier. In response, we have revised the manuscript to include a more thorough comparison with existing methods. Specifically, we have highlighted how our proposed ConvLSTM-TFN network improves upon or differs from previous approaches such as DRN [17][15] and CCR-Net [23][16], particularly in terms of performance, accuracy, efficiency, and robustness. These comparisons are now clearly presented in Section 4.3, which emphasizes the advancements our work brings to the field of channel coding blind recognition. This change can be found in the revised manuscript on page 12, Sec. 4.4, lines 410-420. ---Change in Manuscript--- In our experiments, we compared the proposed ConvLSTM-TFN network against DRN [17]14, CCR-Net [26]15, 1-D CNN, and TextCNN. Among these networks, DRN and CCR-Net are specifically designed for channel coding recognition tasks, demonstrating exceptional performance in this domain and serving as representative works among existing methods. In contrast, 1-D CNN and TextCNN are general-purpose networks with a wide range of applications. A comparison with these networks effectively highlights the advanced capabilities of our proposed network in the realm of channel coding blind recognition. The dataset used for evaluation is the one generated in Section 3.2, along with a hard-decision counterpart generated under identical conditions. Each sample in the dataset contains a noisy version of the input signal as well as the corresponding coding information.
|
|||||||||||||||||||||||||||||||||||||||||||||||||||||
|
Comments 9: The abstract states that the method achieves over 90% recognition accuracy across 17 convolutional code types, with an average accuracy of 98.7%, but I could not find any justification or supporting evidence for this claim in the results section. Please ensure the results section provides detailed data or analysis to validate this statement. |
|||||||||||||||||||||||||||||||||||||||||||||||||||||
|
Response 9: Thank you for your valuable feedback. We sincerely apologize for not clearly pointing out the detailed data or analysis in the results section and conclusion to support the claim of achieving over 90% recognition accuracy across 17 convolutional code types, with an average accuracy of 98.7%. We realize this oversight and the confusion it may have caused. To address this, we have now updated the results section to include the relevant data and analysis that validate this statement. We hope this revision clarifies the matter, and we sincerely appreciate your understanding. This change can be found in the revised manuscript on page 12-13 and 15, Sec. 4.4 and 5, lines 425-437 and 501-510. ---Change in Manuscript--- Page 12-13, Sec.4.4, lines 425-437 Figure 6 (a) depicts the accuracy curves of each network under different decision schemes and varying SNR conditions. When the SNR > -2 dB, our network significantly outperforms all other models, demonstrating its robustness and efficiency. Although CCR-Net exhibits slightly superior performance than our network in extremely low-SNR conditions, resulting in OA 1.86% higher than ConvLSTM-TFN, it fails to maintain this advantage across the entire SNR range. This is also evident in the median line of the box plot presented in Figure 6 (b), where ConvLSTM-TFN demonstrates the best performance in convolutional code blind recognition. Statistical analysis indicates that ConvLSTM-TFN achieves an accuracy exceeding 90% under practical channel conditions with an SNR above 0 dB. Within the SNR range of 0-20 dB, the average blind recognition accuracy of ConvLSTM-TFN reaches an impressive 98.7%. In contrast, our network excels under higher-SNR conditions, significantly outperforming other models and highlighting its superior capability in handling complex environments. Page 15, Sec. 5, lines 501-510 In this study, we propose a novel channel coding recognition network, referred to as ConvLSTM-TFN. This method integrates convolutional layers, LSTM networks, and a self-attention mechanism to effectively capture and distinguish the characteristics of various channel coding types. The hybrid model addresses limitations associated with single-component approaches and demonstrates superior performance by utilizing soft-decision sequences. The model demonstrates a blind recognition accuracy for convolutional codes exceeding 90% when the SNR is greater than 0 dB. Furthermore, it achieves an average accuracy of 98.7% within the SNR range of 0 to 20 dB. Experimental results demonstrate that the superiority of our network compared to existing models, while significantly reducing inter-type confusion.
|
|||||||||||||||||||||||||||||||||||||||||||||||||||||
|
4. Additional clarifications |
|||||||||||||||||||||||||||||||||||||||||||||||||||||
|
To better support the model design and optimization strategies, we have added several references that detail the theoretical background and applications of the ReLU activation function [22], cross-entropy loss [23], and the Adam optimizer [24]. 22. Glorot, X.; Bordes, A.; Bengio, Y. Deep sparse rectifier neural networks. In Proceedings of the fourteenth international conference on artificial intelligence and statistics, Miami, Florida, USA, 11-13 April 2011. 23. Hinton, G.E.; Osindero, S.; Teh, Y.W. A fast learning algorithm for deep belief nets. Neural Computation, 2006, 18, 1527-1554. 24. Kingma, D.P.; Ba, J. Adam: A method for stochastic optimization. In Proceedings of the 3rd International Conference on Learning Representations (ICLR), Vancouver, Canada, 7-9 April 2015.
|
|||||||||||||||||||||||||||||||||||||||||||||||||||||
[1]8. Marazin, M.; Gautier, R.; Burel, G. Algebraic method for blind recovery of punctured convolutional encoders from an erroneous bitstream. IET Signal Processing, 2012, 6, 122-131.
- Ramabadran, S.; Madhukumar, A.S.; Teck, N.W.; See, C.M.S. Parameter estimation of convolutional and helical interleavers in a noisy environment. IEEE Access, 2017, 5, 6151-6167.
[2]10. Moosavi, R.; Larsson, E.G. A fast scheme for blind identification of channel codes, 2011 IEEE Global Telecommunications Conference - GLOBECOM 2011, USA, 05-09 December 2011.
[3]11. Moosavi, R.; Larsson, E.G. Fast blind recognition of channel codes. IEEE Transactions on Communications. 2014, 62, 1393–1405.
[4]18. LeCun, Y.; Boser, B.; Denker, J.S.; Henderson, D.; Howard R.E.; Hubbard, W.; Jakel, L.D. Backpropagation Applied to Handwritten Zip Code Recognition. Neural Computation, 1989, 1, 541-551.
[5]19. Hochreiter, S.; Schmidhuber, J. Long Short-Term Memory. Neural Computation, 1997, 9, 1735-1780.
[6]20. Vaswani, A.; Shazeer, N.; Parmar, N.; Uszkoreit, J.; Jones, L.; Gomez, A.N.; Kaiser, Ł.; Polosukhin, I. Attention Is All You Need. Advances in Neural Information Processing Systems, USA, 4-9 December 2017.
[7]8. Marazin, M.; Gautier, R.; Burel, G. Algebraic method for blind recovery of punctured convolutional encoders from an erroneous bitstream. IET Signal Processing, 2012, 6, 122-131.
[8]9. Ramabadran, S.; Madhukumar, A.S.; Teck, N.W.; See, C.M.S. Parameter estimation of convolutional and helical interleavers in a noisy environment. IEEE Access, 2017, 5, 6151-6167.
[9]10. Moosavi, R.; Larsson, E.G. A fast scheme for blind identification of channel codes, 2011 IEEE Global Telecommunications Conference - GLOBECOM 2011, USA, 05-09 December 2011.
[10]11. Moosavi, R.; Larsson, E.G. Fast blind recognition of channel codes. IEEE Transactions on Communications. 2014, 62, 1393–1405.
[11]15. Qin, X.; Peng, S.; Yang, X.; Yao, Y.D. Deep learning based channel code recognition using TextCNN. 2019 IEEE International Symposium on Dynamic Spectrum Access Networks (DySPAN), USA, 11-14 November 2019.
[12]17. Wang, J.; Tang, C.; Huang, H.; Wang, H.; Li, j. Blind identification of convolutional codes based on deep learning. Digital Signal Processing, 2021, 115, 103086.
[13]7. Huang, X.; Sun, S.; Yang, X.; Peng, S. Recognition of channel codes based on BiLSTM-CNN, 2022 31st Wireless and Optical Communications Conference (WOCC), China, 11-12 August 2022.
[14]25. Opitz, J. A Closer Look at Classification Evaluation Metrics and a Critical Reflection of Common Evaluation Practice. Transactions of the Association for Computational Linguistics, 2024, 12, 820-836.
[15]17. Arab, H.; Ghaffari, I.; Evina, R.M.; Tatu, S.O.; Dufour, S. A hybrid LSTM-ResNet deep neural network for noise reduction and classification of V-band receiver signals. IEEE Access, 2022, 10, 14797-14806.
[16]26. Luo X. Exploiting Deep Learning for Channel Codes Recognition. Master, University of Electronic Science and Technology of China, China, 31 May 2023.

Reviewer 2 Report
Comments and Suggestions for Authors
The work proposes ConvLSTM-TFN (Temporal Feature Network), an innovative blind recognition network that integrates convolutional layers, Long Short-Term Memory (LSTM) networks, and a self-attention mechanism. The proposed approach enhances the acquisition of features from soft decision sequence information, leading to improved recognition performance without necessitating prior knowledge of coding parameters, sequence starting positions, or other metadata. However, there are some questions:
1. Introduction section is too concise, it does not fully explain the progress of existing research work.
2. Figure 3 shows the structure of ConvLSTM-TFN. The left side shows the detailed architecture of the network. However, the link between LSTM and the work is too weak to find the contribution to Blind Recognition in “sensors” journal.
3. Relevant literature needs to be cited, such as “A heterogeneous streaming vehicle data access model for diverse IoT sensor monitoring network management”
4. In experiment section, there are no experiment results regarding Blind Recognition. We can’t find the Blind Recognition result in actual experiment not just performance testing.
5. The tests in Table 5 are not enough. 48.12% and 45.84% are very close, with no significant difference. there are maybe random errors. So the performance improvement lacks persuasiveness.
6. Grammar:
(1) “SNR” in line 71 should be given the full name in first appearance.
(2) Therea are many independent paragraphs that need to be merged such as sentence 243 to 245.
(3) The process from sentence 283 to 290 should be given in detail.
(4)
Comments on the Quality of English Language
The English grammars should be revised.
Author Response
|
Comments 1: Introduction section is too concise, it does not fully explain the progress of existing research work. |
||||||||||||||||||||||||||||||||||||||||
|
Response 1: Thank you for your valuable feedback. We greatly appreciate this suggestion and fully agree with it. In response, we have made an effort to more explicitly highlight the research gaps in the revised manuscript, clearly identifying the limitations of previous studies. This change can be found in the revised manuscript on page 2-3, Sec. 1, paragraph 3-5, lines 46-84 and 100. ---Change in Manuscript--- Convolutional codes typically possess good error-correction capabilities, but error propagation may occur if the decoder incorrectly selects a wrong code or decoding path. The complexities highlighted here emphasize the significance of precise recognition of convolutional codes. Convolutional code recognition and parameter estimation methods have predominantly relied on the Galois field GF(2). Hard decision-based convolutional code recognition methods have been well-developed, enabling the recognition of various convolutional codes without necessitating prior knowledge [8-9][1]. Nevertheless, these approaches generally demonstrate limited resilience to noise. Due to the suboptimal performance of convolutional code recognition under low signal to noise ratio (SNR), there has been a shift towards parameter identification of convolutional codes using soft decision signals, which is currently receiving more attention. The analysis algorithm proposed in reference [10][2] utilizes soft bit information and investigates its efficacy on standard convolutional codes. However, the coding rates studied in the paper are somewhat limited, making it challenging to handle complex real-world environments. Reference [11][3] presents a blind recognition algorithm that utilizes soft information from the received sequence to estimate the posterior probability of syndromes. With the rapid expansion of real-time data and the growing necessity for immediate decision-making, there has been an increased demand for efficient management and access to diverse, heterogeneous data [12][4]. The advent of deep learning has provided new opportunities to address this demand, enabling the automation and optimization of various tasks in signal processing. For instance, it has been used to design adaptive filters [13][5] and automatic noise reduction [14][6], enhancing the robustness of communication systems. In channel coding, deep learning can perform channel code recognition by leveraging the information obtained from the demodulation output. Various deep neural networks, such as TextCNN [15,16][7], have been proposed for channel code recognition. The types of recognition, however, are significantly limited, and the accuracy is relatively low, particularly under low SNR condition. Additionally, a deep learning method based on deep residual networks [17][8] has been proposed for blind recognition of convolutional code parameters from a given soft decision sequence. However, this method fails to consider the possibility that the received encoded sequence may not start from the beginning of a complete codeword, which could impact the model's recognition performance, especially in the presence of channel noise or other disturbances. To mitigate the issue of low accuracy associated with single-type neural networks, a novel channel recognition algorithm based on bi-directional long short-term memory (BiLSTM) and convolutional neural networks (CNN) has been proposed [7][9]. However, it solely discriminates among specific types of convolutional codes, LDPC codes, and polar codes. Table 1 summarizes the strengths and weaknesses of various methods. Table 1. Comparison of Methods: Code Types, Prior Knowledge, and Noise Resilience.
|
||||||||||||||||||||||||||||||||||||||||
|
Comments 2: Figure 3 shows the structure of ConvLSTM-TFN. The left side shows the detailed architecture of the network. However, the link between LSTM and the work is too weak to find the contribution to Blind Recognition in “sensors” journal. |
||||||||||||||||||||||||||||||||||||||||
|
Response 2: Thank you for your thoughtful and constructive feedback. We apologize for any confusion caused by the presentation of the link between LSTM and the blind recognition task. We would like to clarify that the entire ConvLSTM-TFN network is specifically designed for blind recognition tasks, and the LSTM layer, in particular, plays a crucial role in capturing temporal dependencies, which are essential for accurate recognition in noisy environments. In response to your comment, we have strengthened the explanation in the manuscript to more clearly highlight how the network, including the LSTM component, directly contributes to the blind recognition process. We believe this revision better emphasizes the relevance and significance of our approach to blind recognition, especially in the context of the Sensors journal. This change can be found in the revised manuscript on page 5-6, Sec. 3.1, lines 166-178, 191-208. ---Change in Manuscript--- Page 5, Sec. 3.1, lines 166-178: Convolutional codes exhibit long-term dependencies, in contrast to other channel codes, such as linear block codes, only exhibit block-by-block dependencies [21][17]. This dependency is manifested during the encoding process, where each output codeword is influenced not only by the current input symbol but also by preceding input symbols. Therefore, to enhance the effectiveness of blind recognition of convolutional code parameters, it necessitates a deep learning network that is capable of handling long-term dependencies in sequences, such as Long Short-Term Memory (LSTM). Moreover, convolutional codes showcase local correlation and structure through their ability to convert input bits into multiple output bits at specific coding rates. Consequently, we have incorporated convolutional layers for feature extraction, enabling effective capture of these short-term features for more accurate recognition of the patterns and structures of convolutional codes. Simultaneously, we have added a self-attention mechanism to enhance the overall recognition performance of the model. Page 5-6, Sec. 3.1, lines 191-208: The network uses the ReLU activation function in all layers except the output, where a softmax activation is applied to predict blind recognition class probabilities. Cross-entropy loss[23][18] is used as the objective function to optimize the model, while an Adam optimizer[24][19] with learning rate scheduling ensures stable training. Regularization techniques, including dropout and batch normalization, are employed to improve generalization and stabilize learning. In summary, our proposed ConvLSTM-TFN network integrates LSTM, CNN, and self-attention mechanisms, leveraging the advantages of CNNs for local feature extraction, LSTMs for capturing long-term dependencies in data sequences, and self-attention mechanisms for establishing global dependencies. This synergy expands the receptive field, with the objective of improving performance in channel coding blind recognition.
Figure 3. Structure of ConvLSTM-TFN. The left side shows the detailed architecture of the network, while the right side presents the network flowchart. The colors in the flowchart correspond to different components of the detailed architecture, with the yellow section indicating a part that is not explicitly depicted in the detailed architecture on the left. Figure 3 shows the overall network framework of the convolutional code blind recognition, illustrating the key modules and data flow between them.
|
||||||||||||||||||||||||||||||||||||||||
|
Comments 3: Relevant literature needs to be cited, such as “A heterogeneous streaming vehicle data access model for diverse IoT sensor monitoring network management”. |
||||||||||||||||||||||||||||||||||||||||
|
Response 3: Thank you for your valuable feedback. We apologize for not including this relevant reference earlier. We have read the article “A heterogeneous streaming vehicle data access model for diverse IoT sensor monitoring network management” and found it to be an excellent contribution to the field. Its insights into managing heterogeneous data streams are valuable and serve as an example for other domains. In response to your suggestion, we have now incorporated the reference into the revised manuscript. This change can be found in the revised manuscript on page 2, Sec. 1, lines 64-84. ---Change in Manuscript--- With the rapid expansion of real-time data and the growing necessity for immediate decision-making, there has been an increased demand for efficient management and access to diverse, heterogeneous data [12][20]. The advent of deep learning has provided new opportunities to address this demand, enabling the automation and optimization of various tasks in signal processing. For instance, it has been used to design adaptive filters [13][21] and automatic noise reduction [14][22], enhancing the robustness of communication systems. In channel coding, deep learning can perform channel code recognition by leveraging the information obtained from the demodulation output. Various deep neural networks, such as TextCNN [15,16][23], have been proposed for channel code recognition. The types of recognition, however, are significantly limited, and the accuracy is relatively low, particularly under low SNR condition. Additionally, a deep learning method based on deep residual networks [17][24] has been proposed for blind recognition of convolutional code parameters from a given soft decision sequence. However, this method fails to consider the possibility that the received encoded sequence may not start from the beginning of a complete codeword, which could impact the model's recognition performance, especially in the presence of channel noise or other disturbances. To mitigate the issue of low accuracy associated with single-type neural networks, a novel channel recognition algorithm based on bi-directional long short-term memory (BiLSTM) and convolutional neural networks (CNN) has been proposed [7][25]. However, it solely discriminates among specific types of convolutional codes, LDPC codes, and polar codes. Table 1 summarizes the strengths and weaknesses of various methods. |
||||||||||||||||||||||||||||||||||||||||
|
Comments 4: In experiment section, there are no experiment results regarding Blind Recognition. We can’t find the Blind Recognition result in actual experiment not just performance testing. |
||||||||||||||||||||||||||||||||||||||||
|
Response 4: Thank you for your feedback. We apologize for any confusion caused. The focus of our study is indeed on Blind Recognition, and the ConvLSTM-TFN network is specifically designed for this task. The experiment results presented in the manuscript are directly related to Blind Recognition, though we understand that it might not have been clearly emphasized. In the revised manuscript, we have made sure to highlight these points more clearly to ensure that the Blind Recognition aspect of our work is evident. This change can be found in the revised manuscript on page 12-13, Sec. 4.4, lines 421-437. ---Change in Manuscript---
Figure 6. Performance comparison of various deep learning models in channel code recognition. (a) illustrates the recognition accuracy comparison among different methods versus SNR; (b) presents the box plots for each model. Figure 6 (a) depicts the accuracy curves of each network under different decision schemes and varying SNR conditions. When the SNR > -2 dB, our network significantly outperforms all other models, demonstrating its robustness and efficiency. Although CCR-Net exhibits slightly superior performance than our network in extremely low-SNR conditions, resulting in OA 1.86% higher than ConvLSTM-TFN, it fails to maintain this advantage across the entire SNR range. This is also evident in the median line of the box plot presented in Figure 6 (b), where ConvLSTM-TFN demonstrates the best performance in convolutional code blind recognition. Statistical analysis indicates that ConvLSTM-TFN achieves an accuracy exceeding 90% under practical channel conditions with an SNR above 0 dB. Within the SNR range of 0-20 dB, the average blind recognition accuracy of ConvLSTM-TFN reaches an impressive 98.7%. In contrast, our network excels under higher-SNR conditions, significantly outperforming other models and highlighting its superior capability in handling complex environments. |
||||||||||||||||||||||||||||||||||||||||
|
Comments 5: The tests in Table 5 are not enough. 48.12% and 45.84% are very close, with no significant difference. there are maybe random errors. So the performance improvement lacks persuasiveness. |
||||||||||||||||||||||||||||||||||||||||
|
Response 5: Thank you for your valuable feedback, and we apologize for any confusion caused. To provide a more convincing analysis of the advantages of the self-attention mechanism, we have included additional evaluation metrics such as precision and F1-score. These metrics allow for a more comprehensive assessment of the model's performance. Moreover, we have repeated the experiments multiple times to ensure that the results are not due to random errors. This added rigor helps reinforce the persuasiveness of the performance improvements observed. This change can be found in the revised manuscript on pages 9 and 14-15, Sec. 4.1 and 4.5, lines 308-313 and 478-499. ---Change in Manuscript--- Page 9, Sec. 4.1, lines 308-313. For each experiment, we perform five runs and calculate the average of the results. The evaluation metrics include OA (overall accuracy), precision, and the F1 score, which provide a comprehensive assessment of the model's performance across different aspects. These metrics are widely used for performance evaluation and can be computed using standard formulas as outlined in [25][26]. Since our dataset is evenly distributed, OA is equivalent to the recall value. Therefore, we do not include recall as a separate evaluation metric. Page 14-15, Sec. 4.5, lines 478-499. 4.5. Ablation Study Table 7 presents a comparison of the performance between the Baseline model (which solely comprises the backbone network) and the ConvLSTM-TFN model that incorporates a Self-Attention Mechanism. The results indicate that the integration of self-attention consistently enhances all evaluation metrics. Table 7. ConvLSTM-TFN Ablation Study.
Specifically, the OA demonstrates an increase from 64.28% to 65.03%, indicating a modest yet meaningful improvement in overall classification performance. Notably, the most substantial improvements are observed in precision and F1 Score. The precision rises from 61.32% to 66.73%, indicating that the model utilizing self-attention generates fewer false positive predictions and achieves higher reliability in its positive classifications. Similarly, the F1 Score improves from 62.77% to 65.86%, reflecting a more favorable balance between precision and recall in the enhanced model. These results indicate that the Self-Attention mechanism primarily enhances the model's capacity to distinguish complex backgrounds and easily confused categories. The introduction of Self-Attention effectively bolsters feature extraction and sequence modeling, leading to notable improvements in precision and model robustness. Although the overall accuracy improvement is relatively modest, there has been a significant increase in both the model's precision and its overall performance. Consequently, when compared to the baseline model, the ConvLSTM-TFN integrated with the Self-Attention mechanism exhibits superior performance and represents a more effective configuration for modeling tasks. |
||||||||||||||||||||||||||||||||||||||||
|
4. Response to Comments on the Quality of English Language |
||||||||||||||||||||||||||||||||||||||||
|
Point 1: “SNR” in line 71 should be given the full name in first appearance. |
||||||||||||||||||||||||||||||||||||||||
|
Response 1: Thank you for pointing this out. We have now provided the full name of “SNR” (Signal-to-Noise Ratio) at its first appearance in line 54-55. |
||||||||||||||||||||||||||||||||||||||||
|
Point 2: There are many independent paragraphs that need to be merged such as sentence 243 to 245. |
||||||||||||||||||||||||||||||||||||||||
|
Response 2: Thank you for your observation. We agree that some paragraphs can be merged for better readability and flow. In the revised manuscript, we have merged the paragraphs from sentences 243 to 245 (which is now from 250 to 253), as well as other similar sections, to improve the overall structure and coherence of the text. This change has been reflected in the updated version of the manuscript. ---Change in Manuscript--- Page 7, Sec. 3.1.2, lines 250-253: Due to the independence of the characteristics of convolutional codes from the distance between sequence elements, the network incorporates a self-attention mechanism to enhance the model’s ability to capture global information by expanding the receptive field. Page 9, Sec. 4.2.1, lines 316-318: The length of input samples plays a critical role in the blind recognition of convolutional codes, as increasing the sequence length significantly raises both computational complexity and training time. Page 8, Sec. 3.1.2, lines 261-263: Calculating Attention Scores: The attention scores are computed using three learned linear transformations: the Query (Q), Key (K) , and Value (V) : (8) here, WQ, WK, WV are weight matrices. Page 8, Sec. 3.1.2, lines 264-267: Attention Weights: The attention weights are obtained by applying the softmax function to the scaled dot product of the Query and Key matrices:
where dk is the dimensionality of the Key vectors, which helps to stabilize gradients during training. |
||||||||||||||||||||||||||||||||||||||||
|
Point 3: The process from sentence 283 to 290 should be given in detail. |
||||||||||||||||||||||||||||||||||||||||
|
Response 3: Thank you for your valuable feedback. We agree that this section should be more detailed. In response to your comment, we have revised the manuscript to provide a more thorough explanation of the process described from sentences 283 to 290 (which is now from 289 to 301). This additional detail aims to make the methodology clearer and more comprehensible. ---Change in Manuscript--- Page 9, Sec. 3.2, lines 289-301: The process of sample generation is as follows: 1. Generate a binary random sequence of length 300 bits. 2. Perform the corresponding convolutional encoding based on the parameters in Table 3. 3. Modulate the encoded sequence using BPSK modulation to generate the modulated signal. 4. Add AWGN to generate the noisy signal, simulating the effects of channel transmission. 5. Demodulate the signal to obtain the soft decision results corrupted by interference. 6. Randomly select a starting point from the first 0 to 20 numbers in the sequence and extract 256 bits as a sample sequence. 7. Repeat steps 1-6 to generate the complete dataset. |
||||||||||||||||||||||||||||||||||||||||
|
5. Additional clarifications |
||||||||||||||||||||||||||||||||||||||||
|
To better support the model design and optimization strategies, we have added several references that detail the theoretical background and applications of the ReLU activation function [22], cross-entropy loss [23], and the Adam optimizer [24]. 22. Glorot, X.; Bordes, A.; Bengio, Y. Deep sparse rectifier neural networks. In Proceedings of the fourteenth international conference on artificial intelligence and statistics, Miami, Florida, USA, 11-13 April 2011. 23. Hinton, G.E.; Osindero, S.; Teh, Y.W. A fast learning algorithm for deep belief nets. Neural Computation, 2006, 18, 1527-1554. 24. Kingma, D.P.; Ba, J. Adam: A method for stochastic optimization. In Proceedings of the 3rd International Conference on Learning Representations (ICLR), Vancouver, Canada, 7-9 April 2015.
|
||||||||||||||||||||||||||||||||||||||||
[1]8. Marazin, M.; Gautier, R.; Burel, G. Algebraic method for blind recovery of punctured convolutional encoders from an erroneous bitstream. IET Signal Processing, 2012, 6, 122-131.
- Ramabadran, S.; Madhukumar, A.S.; Teck, N.W.; See, C.M.S. Parameter estimation of convolutional and helical interleavers in a noisy environment. IEEE Access, 2017, 5, 6151-6167.
[2]10. Moosavi, R.; Larsson, E.G. A fast scheme for blind identification of channel codes, 2011 IEEE Global Telecommunications Conference - GLOBECOM 2011, USA, 05-09 December 2011.
[3]11. Moosavi, R.; Larsson, E.G. Fast blind recognition of channel codes. IEEE Transactions on Communications. 2014, 62, 1393–1405.
[4]12. Zhou, L.; Tu, W.; Li, Q.; Guan, D. A Heterogeneous Streaming Vehicle Data Access Model for Diverse IoT Sensor Monitoring Network Management. IEEE Internet of Things Journal, 2024, 11, 26929 – 26943.
[5]13. Qian, H.; Yin, G.; Zhang, Q. Deep filtering with adaptive learning rates. IEEE Transactions on Automatic Control, 2022, 68, 3285-3299.
[6]14. Arab, H.; Ghaffari, I.; Evina, R.M.; Tatu, S.O.; Dufour, S. A hybrid LSTM-ResNet deep neural network for noise reduction and classification of V-band receiver signals. IEEE Access, 2022, 10, 14797-14806.
[7]15. Qin, X.; Peng, S.; Yang, X.; Yao, Y.D. Deep learning based channel code recognition using TextCNN. 2019 IEEE International Symposium on Dynamic Spectrum Access Networks (DySPAN), USA, 11-14 November 2019.
- Ni, Y.; Peng, S.; Zhou, L.; Yang, X. Blind identification of LDPC code based on deep learning. 2019 6th International Conference on Dependable Systems and Their Applications (DSA), China, 3-6 January 2020.
[8]17. Wang, J.; Tang, C.; Huang, H.; Wang, H.; Li, j. Blind identification of convolutional codes based on deep learning. Digital Signal Processing, 2021, 115, 103086.
[9] Huang, X.; Sun, S.; Yang, X.; Peng, S. Recognition of channel codes based on BiLSTM-CNN, 2022 31st Wireless and Optical Communications Conference (WOCC), China, 11-12 August 2022.
[10]8. Marazin, M.; Gautier, R.; Burel, G. Algebraic method for blind recovery of punctured convolutional encoders from an erroneous bitstream. IET Signal Processing, 2012, 6, 122-131.
[11]9. Ramabadran, S.; Madhukumar, A.S.; Teck, N.W.; See, C.M.S. Parameter estimation of convolutional and helical interleavers in a noisy environment. IEEE Access, 2017, 5, 6151-6167.
[12]10. Moosavi, R.; Larsson, E.G. A fast scheme for blind identification of channel codes, 2011 IEEE Global Telecommunications Conference - GLOBECOM 2011, USA, 05-09 December 2011.
[13]11. Moosavi, R.; Larsson, E.G. Fast blind recognition of channel codes. IEEE Transactions on Communications. 2014, 62, 1393–1405.
[14]15. Qin, X.; Peng, S.; Yang, X.; Yao, Y.D. Deep learning based channel code recognition using TextCNN. 2019 IEEE International Symposium on Dynamic Spectrum Access Networks (DySPAN), USA, 11-14 November 2019.
[15]17. Wang, J.; Tang, C.; Huang, H.; Wang, H.; Li, j. Blind identification of convolutional codes based on deep learning. Digital Signal Processing, 2021, 115, 103086.
[16]7. Huang, X.; Sun, S.; Yang, X.; Peng, S. Recognition of channel codes based on BiLSTM-CNN, 2022 31st Wireless and Optical Communications Conference (WOCC), China, 11-12 August 2022.
[17]21. Yardi, A.; Kancharla, V.K.; Mishra A. Detecting Linear Block Codes via Deep Learning. 2023 IEEE Wireless Communications and Networking Conference (WCNC), United Kingdom, 26-29 March 2023.
[18]23. Hinton, G.E.; Osindero, S.; Teh, Y.W. A fast learning algorithm for deep belief nets. Neural Computation, 2006, 18, 1527-1554.
[19]24. Kingma, D.P.; Ba, J. Adam: A method for stochastic optimization. In Proceedings of the 3rd International Conference on Learning Representations (ICLR), Vancouver, Canada, 7-9 April 2015.
[20]12. Zhou, L.; Tu, W.; Li, Q.; Guan, D. A Heterogeneous Streaming Vehicle Data Access Model for Diverse IoT Sensor Monitoring Network Management. IEEE Internet of Things Journal, 2024, 11, 26929 – 26943.
[21]13. Qian, H.; Yin, G.; Zhang, Q. Deep filtering with adaptive learning rates. IEEE Transactions on Automatic Control, 2022, 68, 3285-3299.
[22]14. Arab, H.; Ghaffari, I.; Evina, R.M.; Tatu, S.O.; Dufour, S. A hybrid LSTM-ResNet deep neural network for noise reduction and classification of V-band receiver signals. IEEE Access, 2022, 10, 14797-14806.
[23]15. Qin, X.; Peng, S.; Yang, X.; Yao, Y.D. Deep learning based channel code recognition using TextCNN. 2019 IEEE International Symposium on Dynamic Spectrum Access Networks (DySPAN), USA, 11-14 November 2019.
- Ni, Y.; Peng, S.; Zhou, L.; Yang, X. Blind identification of LDPC code based on deep learning. 2019 6th International Conference on Dependable Systems and Their Applications (DSA), China, 3-6 January 2020.
[24]17. Wang, J.; Tang, C.; Huang, H.; Wang, H.; Li, j. Blind identification of convolutional codes based on deep learning. Digital Signal Processing, 2021, 115, 103086.
[25]7. Huang, X.; Sun, S.; Yang, X.; Peng, S. Recognition of channel codes based on BiLSTM-CNN, 2022 31st Wireless and Optical Communications Conference (WOCC), China, 11-12 August 2022.
[26]25. Opitz, J. A Closer Look at Classification Evaluation Metrics and a Critical Reflection of Common Evaluation Practice. Transactions of the Association for Computational Linguistics, 2024, 12, 820-836.

Round 2
Reviewer 1 Report
Comments and Suggestions for Authors
No further comments.
Reviewer 2 Report
Comments and Suggestions for Authors
The study can be accepted in present form.